# IL-37 Gene and Cholesterol Metabolism: Association of Polymorphisms with the Presence of Hypercholesterolemia and Cardiovascular Risk Factors. The GEA Mexican Study

**DOI:** 10.3390/biom10101409

**Published:** 2020-10-05

**Authors:** Fabiola López-Bautista, Rosalinda Posadas-Sánchez, Christian Vázquez-Vázquez, José Manuel Fragoso, José Manuel Rodríguez-Pérez, Gilberto Vargas-Alarcón

**Affiliations:** 1Department of Molecular Biology, Instituto Nacional de Cardiología Ignacio Chávez, Mexico City 14080, Mexico; nutrifabs@gmail.com (F.L.-B.); trascan@hotmail.com (C.V.-V.); mfragoso1275@yahoo.com.mx (J.M.F.); josemanuel_rodriguezperez@yahoo.com.mx (J.M.R.-P.); 2Department of Endocrinology, Instituto Nacional de Cardiología Ignacio Chávez, Mexico City 14080, Mexico; rossy_posadas_s@yahoo.it

**Keywords:** cardiovascular risk factors, hypercholesterolemia, inflammation, interleukin 37, polymorphisms

## Abstract

Interleukin 37 (*IL-37*) is an anti-inflammatory cytokine involved in the regulation of cholesterol homeostasis, reducing the levels of plasma cholesterol, fatty acids, and triglycerides. The aim of the present study was to evaluate the association of the IL-37 polymorphisms with the presence of hypercholesterolemia (HC), and with cardiovascular risk factors. Nine IL-37 polymorphisms (rs2708965, rs2708962, rs6717710, rs2708961, rs2708960, rs2708958, rs2723187, rs2708947, and rs2723192) were determined by TaqMan assays in a group of 1292 individuals (514 with and 778 without hypercholesterolemia) belonging to the cohort of the GEA Mexican Study. The associations were evaluated by logistic regression, using inheritance models adjusted by confounding variables. Under codominant 1 model, the rs2708961 (OR = 0.51, *p* = 0.02), rs2723187 (OR = 0.35, *p* = 0.005), and rs2708947 (OR = 0.49, *p* = 0.02) polymorphisms were associated with low risk of HC. The association of the polymorphisms with cardiovascular risk factors was evaluated independently in HC and non-HC individuals. In non-HC individuals, some polymorphisms were associated with the risk of having high levels of LDL-C, glucose, and high risk of T2DM, and low risk of having high visceral abdominal fat. On the other hand, in individuals with HC five, polymorphisms were associated with high levels of C-reactive protein. The *IL-37* rs2708961, rs2723187, rs2708947 polymorphisms were associated with low risk of HC, and some *IL-37* polymorphisms were associated with cardiometabolic factors in both individuals with and without HC.

## 1. Introduction

Coronary artery disease (CAD) is a chronic, progressive, and multifactorial disease modulated by genetic and environmental factors [1]. This disease is a consequence of the presence of a long period of atherosclerosis, which, in the last few decades, has been considered an inflammatory disease [2]. Macrophages play an important role in the development of atherosclerosis. Chronic inflammation and the dysregulation of cholesterol metabolism by macrophages within the atherosclerotic plaque are the main phenomena involved in the onset and progression of atherosclerosis [3]. The increased uptake of oxidized low-density lipoprotein (oxLDL) and/ or reduced cholesterol efflux leads to the deposition of esterified cholesterol in the cytoplasm of macrophages and the generation of foam cells [4]. In the same way, the intake of ox-LDL by macrophages [5] leads to the activation of pattern recognition receptors and TLR expression, which promotes the production of immune-inflammatory mediators [6,7]. Interleukin 37 is a cytokine that belongs to the interleukin 1 family and is produced in an important way by macrophages, epithelial cells, dendritic cells, and T cells [8]. In macrophages and dendritic cells, IL-37 dramatically reduces the secretion of pro-inflammatory cytokines and limits the activation and differentiation of macrophages [9]. It has been reported that IL-37 expression in foam cells, located in the atherosclerotic plaques [10], has a potential role in the macrophage lipid homeostasis and in the development of atherosclerosis disease. In the same way, the expression of IL-37 is induced by TGFB, and its activity is regulated by the transcription factor Smad3, suggesting a potential role for IL-37 in regulating macrophage cholesterol absorption. The role of IL-37 in the regulation of cholesterol homeostasis was demonstrated in a transgenic mouse model for IL-37. These mice were fed a high-fat diet for 16 weeks, showing reduced levels of plasma cholesterol, fatty acids, and triglycerides when compared to wild mice [11]. These researchers showed that the mechanism by which IL-37 could be participating in in the macrophage lipid-handling involves AMP-activated kinase (AMPK), an important signaling protein downstream of IL-37. In this context, AMPK activation could attenuate the accumulation of ox-LDL in macrophages through the regulation of the cholesterol flux mediators of these cells, such as ABCA1 and ABCG1 [12]. In previous work, Yin et al. reported the association of the IL-37 rs3811047 polymorphism with CAD and with a decreased mRNA expression level of IL-37 [13]. Thus, the aim of the present study was to evaluate the association of the IL-37 polymorphisms with the presence of hypercholesterolemia, and with cardiovascular risk factors in a cohort of Mexican individuals. Based on a bioinformatic analysis, we selected the rs2708965, rs2708962, rs6717710, rs2708961, rs2708960, rs2708958, rs2723187, rs2708947, and rs2723192 polymorphisms for the present study.

## 2. Materials and Methods 

### 2.1. Study Population

This report is a cross-sectional analysis of the baseline evaluation of the GEA study control group and included 1292 individuals (514 with and 778 without hypercholesterolemia). These individuals were healthy and asymptomatic, without a family history of premature CAD, recruited from blood bank donors and through brochures posted in social service centers. Exclusion criteria included congestive heart failure and liver, renal, thyroid, or oncological disease. Standardized questionnaires were applied to all participants to obtain demographic information, history of nutritional habits, physical activity, family medical history, alcohol consumption, and pharmacological treatment. Demographic, clinical, anthropometric, biochemical parameters, and cardiovascular risk factors were evaluated in all individuals, as previously described [14,15,16]. Briefly, anthropometric parameters, including waist circumference and body mass index (BMI), were calculated as weight in kilograms divided by height in square meters. Current smoking was considered when individuals self-reported the current use of cigarettes. Obesity was defined as BMI ≥ 30 kg/m^2^. Hypoalphalipoproteinemia, hypertriglyceridemia, and elevated LDL-C were defined using the criteria from the American Heart Association’s National Heart, Lung, and Blood Institute Scientific Statement on the MS [17]. Central obesity was considered when waist circumference was 90 cm in men and 80 cm in women [18]. Type 2 diabetes mellitus (T2DM) was defined by the American Diabetes Association criteria, with a fasting glucose ≥ 126 mg/dL and was also considered when participants reported glucose-lowering treatment or a physician’s diagnosis of diabetes. Insulin resistance was estimated using the homeostasis model assessment (HOMA-IR) and the presence of insulin resistance was considered when values were ≥75th percentile (3.58 in women and 3.12 in men). Hypertension was defined as systolic blood pressure ≥ 140 mmHg and/or diastolic blood pressure ≥ 90 mmHg, or the use of oral antihypertensive therapy. Hypercholesterolemia (HC) was defined as levels of total-cholesterol > 200 mg/dL. Elevated high sensitivity C-reactive protein (hs-CRP) was considered when its values were ≥ 3 mg/L. Increase in visceral abdominal fat (VAF) was defined as VAF ≥ 75th percentile (122.0 cm^2^ in women and 151.5 cm^2^ in men). These cutoff points were obtained from a GEA Mexican study sample of 131 men and 185 women without obesity and with normal values of blood pressure, fasting glucose, and lipids. Computed tomography of the chest and abdomen was performed on all subjects using a 64-channel multidetector helical computed tomography system (Somatom Cardiac Sensation, 64, Forcheim, Germany). Total, subcutaneous, and visceral abdominal fat (TAF, SAF, VAF) areas were quantified as described by Kvist et al. [19].

All participants provided informed consent. The study was conducted in accordance with the Helsinki Declaration.

### 2.2. Genetic Analysis

High-molecular-weight genomic DNA was extracted from peripheral blood using the QIAamp DNA Blood Mini kit (QIAGEN, Hilden, Germany). According to the manufacturer’s instructions (Applied Biosystems, Foster City, CA, USA), rs2708965, rs2708962, rs6717710, rs2708961, rs2708960, rs2708958, rs2723187, rs2708947, and rs2723192 *IL-37* polymorphisms were determined using 5′ exonuclease TaqMan genotyping assays on an ABI Prism 7900HT Fast Real-Time PCR system. In order to corroborate the adequate assignment of the genotypes in the TaqMan assays, we randomly selected and repeated 10% of the samples. These samples were 100% concordant in two independent assays. 

### 2.3. Functional Prediction Analysis 

To anticipate the probable effect of the *IL-37* polymorphisms, we utilized the following bioinformatics tools: FastSNP [20], Human-transcriptome Database for Alter- native Splicing [21], SNP Function Prediction [22], HSF [23], SNPs3D [24], Splice Port: An Interactive Splice Site Analysis Tool [25], and ESE finder [26]. After this bioinformatic analysis, we selected for the study those polymorphisms with possible functional effect (create binding sites for transcription factors for those polymorphisms located in the promoter region and change of amino acid in those polymorphisms located in the exons).

### 2.4. Statistical Analysis

Data are expressed as the mean (standard deviation), median (interquartile range), or frequencies. Analysis of continuous and categorical variables was made using Student t, Mann–Whitney U and chi-square tests, as appropriate. Associations of polymorphisms with HC and cardiovascular risk factors under codominant 1, codominant 2, heterozygote, additive, recessive, and dominant models were evaluated using logistic regression analysis. All models were adjusted by confusing variables. Models were constructed using one variable at a time; final models included variables with biological relevance or with statistical significance or both. All the polymorphisms studied were in Hardy–Weinberg equilibrium (*p* > 0.05). We used STATA/MP: Release 15 (College Station, TX: Stata Corp LLC) software for all analyses. A value of *p* < 0.05 was considered significant. Linkage disequilibrium and construction of haplotypes were made using the Haploview version 4.2 [27] (Broad Institute of Massachusetts Institute of Technology and Harvard University, Cambridge, MA, USA).

## 3. Results

### 3.1. Characteristics of the Study Group

In Table 1, demographic, lifestyle, clinical and biochemical characteristics, and tomographic data of the study population are shown. Blood pressure, triglycerides, total cholesterol, HDL-cholesterol, LDL-Cholesterol, visceral adipose tissue, hypertriglyceridemia, and hs-CRP were higher in HC individuals when compared to non-HC. On the other hand, individuals without HC presented a low hypoalphalipoproteinemia and T2DM. 

### 3.2. Association of the IL-37 Polymorphisms and Haplotypes with HC

Under codominant 1 model, the rs2708961 (OR = 0.51, 95%CI: 0.28-0.93, *p* = 0.02), rs2723187 (OR = 0.35, 95%CI: 0.17–0.73, *p* = 0.005), and rs2708947 (OR = 0.49, 95%CI: 0.26–0.92, *p* = 0.02) polymorphisms were associated with low risk of HC (Table 2).

The nine study polymorphisms were in strong linkage disequilibrium (Figure 1) and one of them was associated with low risk of HC (OR = 0.46, 95%CI: 0.25–0.88, *p* = 0.007) (Table 3).

### 3.3. Association of the IL-37 Polymorphisms with Cardiovascular Risk Factors

The association of the *IL-37* polymorphisms with cardiovascular risk factors was evaluated independently in individuals with and without HC (Figure 2). In individuals without HC, under codominant 1 model, some polymorphisms were associated with high levels of LDL-C (rs2708965, rs2708962, rs2708961, rs2723187, rs2708947, and rs2723192) (Figure 2A), glucose (rs2708965, rs2708962, rs2708961, rs2708960, rs2708958, rs2723187, rs2708947, rs2723192) (Figure 2B), VAF (rs2708965, rs2708961, rs2708960, and rs2708958) (Figure 2C), and with high risk of T2DM (rs2708965, rs2708962, rs2708961, rs2708960, rs2708958, rs2723187, rs2708947, rs2723192) (Figure 2D). On the other hand, in individuals with HC, the rs2708965, rs2708962, rs6717710, rs2708961, and rs2708960 were associated with high levels of hs-CRP (Figure 2E).

## 4. Discussion

Despite the important role of the IL-37 in the regulation of cholesterol homeostasis, no association studies of polymorphisms in the gene that encodes this cytokine with hypercholesterolemia have been reported. However, Yin et al., reported that the rs3811047 polymorphism located in the *IL-37* gene confers a significant risk of coronary artery disease (CAD) in two independent cohorts from China [13]. Studying nine *IL-37* polymorphisms in a cohort of Mexican individuals with and without HC, we found three of them associated with a decreased risk of HC. rs2708961 (located in the promoter), rs2723187 (located in exon 2), and rs2708947 (located in exon 3) showed a reduction in risk of 51%, 35%, and 49%, respectively. The rs3811047 variant previously associated with the risk of CAD was not determined in the present work, because, in our informatic analysis, this variant was not functional and possibly benign. Instead of this variant, we studied the rs6717710 polymorphism, which was functional (creates a binding site for GATA transcription factor), and in the same way, was in complete linkage disequilibrium with the rs3811047 polymorphism. The protective effect of IL-37 for hyperlipidemia has been observed in mice (IL-37-Tg) fed a high-fat diet, which, after 16 weeks of intervention, showed reduced plasma cholesterol levels and decreased free fatty acids and triglycerides compared to wild controls [11,28]. The authors suggest that the reduced serum cholesterol levels in these mice might be an effect of AMP-activated kinase (AMPK) activation in the liver, in response to IL-37. The accumulation of oxidized LDL (ox-LDL) in macrophages of these mice could be attenuated by AMPK through the regulation of the expression of cholesterol flow mediators as the cassette transporter of ATP binding A1 (ABCA1) and G1 (ABCG1) [12]. It is well-known that high levels of cholesterol are a risk factor for atherosclerosis, and recently it was reported that treatment with IL-37 recombinant diminishes the accumulation of lipids and the foam cells’ formation into the atherosclerotic plaque, as regulated by the transcription factor Smad3 [29,30]. In the same way, the nine polymorphisms were in linkage disequilibrium, and one of the formed haplotypes (*ATCCTGTCA*) was associated with a low risk of HC. This haplotype includes the rs2708961 *C*, rs2723187 *T*, and rs2708947 *C* alleles, which were associated with a low risk of HC when the polymorphisms were analyzed independently.

An informatics analysis showed that the rs2708961, located in the promoter, can produce binding sites for the transcription factors BRCA and MYB. These factors could affect the transcription of the cytokine and modify its expression, and, consequentially, the levels of the molecules. On the other hand, the other two polymorphisms associated with low risk of HC are located in exon two (rs2723187) and three (rs2708947), both of which produce a change in amino acid in these positions. The rs2723187 produces a change of Proline to Leucine, both of which are hydrophobic amino acids, whereas rs2708947 produces a change of tryptophan to arginine; tryptophan is a polar neutral amino acid and arginine is a basic amino acid. This last change could have a significant impact on the structure and function of the protein.

In individuals without HC, some polymorphisms were associated with high levels of LDL-C, glucose, and a high risk of T2DM. It is important to notice that three of the six polymorphisms associated with high levels of LDL-C were associated with a low risk of HC. These polymorphisms identify subjects with a lower risk of having high total cholesterol. However, when analyzing the distribution of cholesterol in lipoproteins, the polymorphisms identify individuals with a higher risk of having elevated LDL-C, despite not having total cholesterol levels above 200 mg/dL, a criterion used in this study to define HC. Given this, additional studies are required to confirm our findings, as well as functional studies to define the possible effect of these polymorphisms on lipoprotein metabolism. In addition, the association of polymorphisms with LDL-C levels could be relevant, considering that these polymorphisms could be considered risk markers in individuals who do not meet the criterion of HC considered as elevated total-cholesterol. It has been reported that high IL-37 mRNA levels in adipose tissue correlated positively with insulin sensitivity and with a lower inflammation of this tissue [11]. Recently, Li et al. [31] reported that the IL-37 was highly expressed in the elderly T2DM patients, and that this expression correlated with the insulin resistance index, suggesting the association of this cytokine with insulin resistance and sensitivity. It has been reported that in vitro IL-37 expression reduces macrophage inflammation and migration and inhibits modified LDL uptake, suggesting an athero-protective role for macrophage-expressed IL-37 [32]. 

The link between the adipose tissue distribution and complications in obese individuals was reported many years ago; however, in more recent years, it has been demonstrated that body fat distribution, rather than total amount of fat, is related with obesity-related disorders [33] and it is now well known that visceral abdominal fat is related with metabolic abnormalities. Studying patients with severe obesity, Moschen et al., reported an increased expression of IL-37 in adipose tissue, which, after weight loss, was higher in subcutaneous/visceral adipose tissue compared with their liver expression [34]. The changes in expression of IL-37 and other IL-1 family cytokine members in adipose and liver tissue in patients with weight loss contributed to the improvement of insulin resistance and inflammations in these patients. In our study, we observe an association of four *IL-37* polymorphisms with low levels of VAF in individuals without HC, suggesting a protective role of these polymorphisms for VAF and, as a result, for metabolic abnormalities in these individuals.

Individuals with HC presented levels of hs-CRP moderately higher than individuals without HC, which reflects the inflammatory process present in these subjects. In HC individuals, five IL-37 polymorphisms were associated with high levels of inflammation (hs-CRP ≥ 3 mg/dL). Unfortunately, in this case, we do not know if these polymorphisms were associated with low or high levels of IL-37 in these individuals. This is important because it is well known that IL-37 can reduce the production of proinflammatory cytokines, such as IL-6 [10].

A strength of our study is the inclusion of individuals from the GEA project cohort, who had been evaluated from a clinical, demographic, biochemical, tomographic, and lifestyle point of view. All these variables have been analyzed in the present study, and some of them have been considered as possible confounding factors and used for the adjustment of the models. However, some limitations should be considered. We did not measure IL-37 levels in the individuals included in the study, and so we could not establish if there were differences in these levels in HC and non-HC individuals. In the same way, this limitation prevented us from establishing whether there was an association between the polymorphisms studied and the levels of this cytokine. Finally, we did not carry out experiments that allowed us to establish whether the polymorphisms have a functional consequence, we only used the computer approach.

## 5. Conclusions

In summary, our data suggest an association of the rs2708961, rs2723187, and rs2708947 polymorphisms in the *IL-37* gene with a low risk of HC. A haplotype that includes the nine IL-37 polymorphisms associated with low risk of present HC is also reported. Some *IL-37* polymorphisms were associated with the risk of having high levels of LDL-C, glucose, and high risk of T2DM, and low risk of having high VAF in individuals without HC. In individuals with HC, five polymorphisms were associated with high levels of hs-CRP. To the best of our knowledge, this is the first study that evaluates the association of *IL-37* polymorphisms with HC and cardiovascular risk factors. For this reason, the detected associations are not yet definitive, and replicate studies in independent populations are warranted to confirm these findings.

## Figures and Tables

**Figure 1 biomolecules-10-01409-f001:**
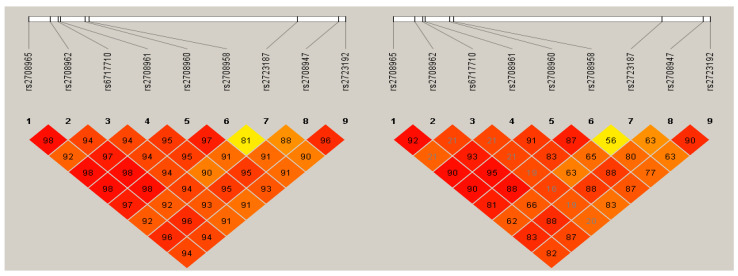
Linkage disequilibrium analysis. Delta (D’) and r^2^ values are shown.

**Figure 2 biomolecules-10-01409-f002:**
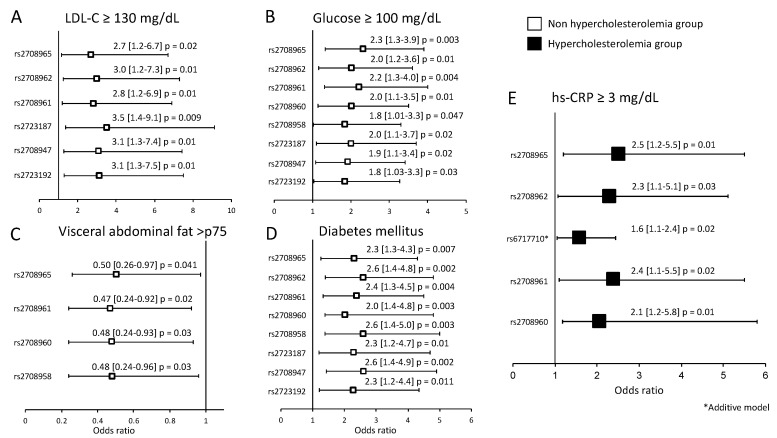
Association of the polymorphisms with metabolic variants. In individuals without HC, six polymorphisms were associated with LDL-C ≥ 130 mg/dL (**A**), eight polymorphisms were associated with glucose ≥ 100 mg/dL (**B**), four polymorphisms were associated with visceral abdominal fat > p75 (**C**), and eight polymorphisms were associated with a high risk of T2DM (**D**). In individuals with HC, five polymorphisms were associated with hs-CRP ≥ 3 mg/dL (**E**). hs-CRP: high sensitive C-reactive protein.

**Table 1 biomolecules-10-01409-t001:** Characteristics of the study groups.

	No HC *n* = 778	HC *n* = 514	*p*
**Demographic**			
Gender (% male)	51.1	48.8	0.41
Age (years)	53 ± 9.8	54.2 ± 8.4	0.02
**Clinical and Biochemical** **Characteristics**			
Body mass index (kg/m^2^)	28.1 ± 4.1	28.5 ± 3.8	0.10
Systolic blood pressure (mmHg)	116 ± 17	118.7 ± 18.2	0.02
Diastolic blood pressure (mmHg)	71 ± 9.2	73.2 ± 9.8	0.001
Triglycerides (mg/dL)	132 (98–184)	175 (134–239)	<0.001
Total cholesterol (mg/dL)	169.5 ± 21.7	227.5 ± 24.6	<0.001
HDL-cholesterol (mg/dL)	44.4 ± 12.6	48.6 ± 13.7	<0.001
LDL-cholesterol (mg/dL)	100.3 ± 20.9	145.9 ± 25.9	<0.001
Glucose (mg/dL)	99.2 ± 34.1	99.8 ± 33.7	0.75
Insulin (μU/mL)	17.2 (12.3–23.8)	17.3 (12.5–23)	0.92
HOMA-IR	3.9 (2.65–5.71)	4.0 (2.7–5.8)	0.59
hs-CRP (mg/L)	1.39 (0.78–2.9)	1.65 (0.85–3.37)	0.16
Visceral adipose tissue (cm^2^)	146 (105–188)	155 (116–204)	0.003
**Lifestyle**			
Physical Activity Index	7.8 ± 1.27	7.8 ± 1.2	0.99
Alcohol, gr/day	0.29 (0.01–1.47)	0.45 (0.09–1.47)	0.01
Smoke, %			
Current	172 (22.1)	111 (21.6)	0.93
Past	290 (37.8)	189 (36.7)	
Never	316 (40.6)	214 (41.6)	
Saturated fat intake (kcal)	226.8 ± 83.2	221.5 ± 85.3	0.27
**Medical History, %**			
Hypoalphalipoproteinemia	56	42	<0.001
Hypertriglyceridemia	40	64	<0.001
LDL-cholesterol ≥ 130 mg/dL	5.1	76	<0.001
Obesity	30.4	32.4	0.13
Type 2 Diabetes Mellitus	15	11.2	0.03
Hypertension	23.5	25.4	0.22
hsCRP ≥ 3 mg/dL	24.6	29.1	0.042
Lipid-lowering therapy	14	20	0.001

Data are shown as mean ± SD, median (interquartile range). Significant value of *p* < 0.05: t-student, U Mann–Whitney, and Chi-square test. HOMA-IR: homeostasis model assessment–insulin resistance; hs-CRP: High sensitivity C reactive protein; HDL: High-density lipoprotein; LDL: Low-density lipoprotein; No HC: No hypercholesterolemia; HC: Hypercholesterolemia.

**Table 2 biomolecules-10-01409-t002:** Association of *IL-37* gene polymorphisms with hypercholesterolemia

Polymorphism	Genotype Frequency			MAF	Model	OR (95% CI)	*p*
rs2708961	*TT*	*TC*	*CC*				
No (*n* = 778)	0.91	0.08	0.001	0.04	Codominant 1	0.51 (0.28–0.93)	0.02
Yes (*n* = 514)	0.94	0.05	0	0.02			
rs2723187	*CC*	*CT*	*TT*				
No (*n* = 778)	0.93	0.06	0.001	0.03	Codominant 1	0.35 (0.17–0.73)	0.005
Yes (*n* = 514)	0.96	0.03	0	0.01			
rs2708947	*TT*	*TC*	*CC*				
No (*n* = 778)	0.91	0.07	0.001	0.04	Codominant 1	0.49 (0.26–0.92)	0.02
Yes (*n* = 514)	0.94	0.04	0.001	0.02			

MAF = minor allele frequency. EHW > 0.05. Adjusted by age, gender, BMI, lipid-lowering use, kcal saturated fat, physical activity, gr alcohol/day, and visceral adipose tissue. Codominant 1 model (heterozygous vs. major allele homozygous). Only significant polymorphisms and models are shown.

**Table 3 biomolecules-10-01409-t003:** Haplotype frequencies in individuals with and without HC.

Haplotype	Sequence	HC YES	NO	OR (95%CI)	*p*
H1	*C-C-T-T-C-A-C-T-G*	0.874	0.859	1.13 (0.89–1.43)	0.142
H2	*C-C-C-T-C-A-C-T-G*	0.097	0.102	0.93 (0.71–1.22)	0.310
H3	*A-T-C-C-T-G-T-C-A*	0.013	0.027	0.46 (0.25–0.88)	0.007

Position at the chromosome: rs2708965, rs2708962, rs6717710, rs2708961, rs2708960, rs2708958, rs2723187, rs2708947, rs2723192. HC: Hypercholesterolemia

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
