# Peer review of "IL-37 Gene and Cholesterol Metabolism: Association of Polymorphisms with the Presence of Hypercholesterolemia and Cardiovascular Risk Factors. The GEA Mexican Study"

_biomolecules, 2020, doi:10.3390/biom10101409_

Round 1
Reviewer 1 Report
The manuscript by Lopez-Bautista addresses the effects of several polymorphisms associated with the IL-37 gene with cardiovascular disease risk factors. IL-37 is a recently described member of the IL-1 family and has been shown to have anti-inflammatory effects, therefore its regulation and function is of high interest to the cardiovascular community. ~1300 patient samples from the GEA Mexican Study Cohort, included hypercholesterolemic and non-hypercholesterolemic patients, were used in the study. Conclusions suggest that a set of polymorphisms are associated low cardiovascular risk in hypercholesterolemic individuals, whereas other sets of polymorphisms were associated with elevated cardiovascular risk in non-hypercholesterolemic individuals.
Together this paper is well-powered, rigorous, and appropriately addresses the hypothesis. Their conclusions open the possibility of future studies to address the effect of the described SNPs in association with gene expression or functional changes of IL-37. While these additional data would help strengthen the conclusions of the paper, these limitations were adequately addressed in the discussion and are emphasized as the next step in linking the role of IL-37 polymorphisms in cardiovascular risk.
Concerns:
- Please cite and discuss the article by Yin et. al. Scientific Reports 2017 (PMID 28181534), “Genomic variant in IL-37 confers a significant risk of coronary artery disease”. Please also include how the nine SNPs being interrogated in the present manuscript were selected and why the rs3811047 variant was not identified/or pursued in the current study.
Author Response
Answer: As is suggested by the reviewer, the manuscript of Yin et al., has been cited and discussed. The phrase “In previous work, Yin et al., reported the association of the IL-37 rs3811047 polymorphism with CAD and with a decreased mRNA expression level of IL-37 [13].” has been added in the introduction section.
On the other hand, the phrase “However, Yin et al., reported that the rs3811047 polymorphism located in the IL-37gene confers a significant risk of CAD in two independent cohorts from China [13].” has been added in the discussión section.
The reference “Yin, D.; Naji, D.H.; Xia, Y.; Li, S.; Bai, Y.; Jiang, G.; Zhao, Y.; Wang, X.; Huang, Y.; Chen, S.; Fa, J.; Tan, C.; Zhou, M.; Zhou, Y.; Wang L.; Liu Y.; Chen, F.; Liu, J.; Chen, Q.; Tu, X.; Xu, C.; Qing K. Wang. Genomic Variant in IL-37 Confers A Significant Risk of Coronary Artery Disease. Sci Report 2017, 7:42175.” has been added.
In order to explain why the rs3811047 variant was not determined in our study, the phrase “The rs3811047 variant previously associated with the risk of CAD was not determined in the present work due to in our informatic analysis this variant was not functional and possible benign. Instead of this variant, we studied the rs6717710 polymorphism, which was functional (creates a binding site for GATA transcription factor) and in the same way, was in complete linkage disequilibrium with the rs3811047 polymorphism.” has been added in the discussion section.
Reviewer 2 Report
This article shows that the association of three IL-37 polymorphisms was associated with low risk of HC and some polymorphisms was associated with glucose ≥ 100 mg/dL, diabetes, and visceral abdominal fat >p75 in individuals without HCs in a cohort of Mexican individuals. There are few reports on association of polymorphisms in the IL-37 gene and cholesterol metabolism. Thus, this article is interesting, but it's a little confusing. How did the authors select 9 SNPs analyzed based on a bioinformatic analysis in this study? The authors described that six polymorphisms were associated with LDL-C ≥ 130 mg/dL in individuals without HC in Figure 2. Three polymorphisms in 6 polymorphisms were associated with low risk of HC. The mean LDL-C level was 145.9±25.9 mg/dL in patients with HC in Table 1. The reviewer would like to know that how the authors think about the relationship between cholesterol and these polymorphisms in this study. In Table 2, various genetic models were used according to the SNPs. How did the authors select appropriate models? The reviewer thinks that it was better to show the results of the same models for all polymorphisms.
Author Response
1.- How did the authors select 9 SNPs analyzed based on a bioinformatic analysis in this study?
Answer: The SNPs were selected considering the results of the bioinformatic analysis. We included those SNPs with possible functional effects. In order to clarify this point, the phrase “After this bioinformatic analysis, we selected for the study, those polymorphisms with possible functional effect (create binding sites for transcription factors for those polymorphisms located in the promoter region and change of amino acid in those polymorphisms located in the exons).” has been added in the material and methods section.
2.- The authors described that six polymorphisms were associated with LDL-C ≥ 130 mg/dL in individuals without HC in Figure 2. Three polymorphisms in 6 polymorphisms were associated with low risk of HC. The mean LDL-C level was 145.9±25.9 mg/dL in patients with HC in Table 1. The reviewer would like to know that how the authors think about the relationship between cholesterol and these polymorphisms in this study.
Answer: In order to clarify this point, the phrase “It is important to notice that three of the six polymorphisms associated with high levels of LDL-C, were associated with a low risk of HC. These polymorphisms identify subjects with a lower risk of having high total cholesterol. However, when analyzing the distribution of cholesterol in lipoproteins, the polymorphisms identify individuals with a higher risk of having elevated LDL-C despite not having total cholesterol levels above 200 mg/dL, a criterion used in this study to define HC. Given this, additional studies are required to confirm our findings, as well as functional studies to define the possible effect of these polymorphisms on lipoprotein metabolism. In addition, the association of polymorphisms with LDL-C levels could be relevant considering that these polymorphisms could be considered risk markers in individuals who do not meet the criterion of HC considered as elevated total-cholesterol.” has been added in the discussion section.
3.- In Table 2, various genetic models were used according to the SNPs. How did the authors select appropriate models? The reviewer thinks that it was better to show the results of the same models for all polymorphisms.
Answer: We agree with the reviewer, and table 2 has been modified. In the new table, only the codominant 1 model is shown. The corresponding corrections in the text were done.
Round 2
Reviewer 2 Report
The manuscript has been improved by revision. Please check the English grammar and spelling.